

# A reappraisal of *Theroteinus* (Haramiyida, Mammaliaformes) from the Upper Triassic of Saint-Nicolas-de-Port (France)

Maxime Debuysschere

Centre de Recherches sur la Paléobiodiversité et les Paléoenvironnements (CR2P), UMR 7207 CNRS-MNHN-UPMC (SU), Paris, France

## ABSTRACT

The earliest mammaliaforms are difficult to assess because the fossil record is poor and because their distinctive morphologies cannot be directly compared with more recent mammaliaforms. This is especially true for the haramiyid genus *Theroteinus*, only known in the Saint-Nicolas-de-Port locality (Rhaetian, France). This study presents a new definition of the type-species *Theroteinus nikolai*. A new species *Theroteinus rosieriensis*, sp. nov., is named and distinguished by the lingual shift of distal cusps, a larger size, and a stockier occlusal outline. Comparisons with *Eleutherodon*, *Megaconus* and *Millsodon* suggest that *Theroteinus* has potential close relatives among the Jurassic haramiyids.

## INTRODUCTION

The earliest mammaliaforms are notoriously poorly known because of the scarcity of specimens (most often isolated teeth) and the difficulty to assess their relationships with later mammaliaforms (including mammals themselves). Among them, haramiyids have long been considered as a very peculiar group that is difficult to study (*e.g.*, *Simpson, 1928*; *Kielan-Jaworowska, Cifelli & Luo, 2004*). However, even within haramiyids, the genus *Theroteinus Sigogneau-Russell, Frank & Hemmerlé, 1986* is distinctive and has always been set apart. This genus was hitherto known only by a dozen isolated teeth, all from the locality of Saint-Nicolas-de-Port (Rhaetian, north-eastern France), which has yielded a very diversified and abundant micro-vertebrate assemblage (see below). Because of the peculiar morphology of *Theroteinus*, some authors cast doubt on its haramiyidan referral (*Sigogneau-Russell, 1983a*; *Sigogneau-Russell, Frank & Hemmerlé, 1986*) and later separated it from all other haramiyids (*Hahn, Sigogneau-Russell & Wouters, 1989*; *Butler, 2000*; *Hahn & Hahn, 2006*). Recently, several new haramiyids were described which significantly increased the diversity and the disparity of the order (*e.g.*, *Zheng et al., 2013*; *Zhou et al., 2013*; *Bi et al., 2014*). In this study, new *Theroteinus* material is described, which forms the basis for a systematic reassessment and an update of relationships of this genus within haramiyids.

Corresponding author
Maxime Debuysschere,
maxime.debuysschere@edu.mnhn.fr

## HISTORICAL BACKGROUND

In 1983, Sigogneau-Russell described three very peculiar teeth: MNHN.F.SNP 61 W was considered to represent a new haramiyid and MNHN.F.SNP 78 W, and MNHN.F.SNP 2 Ma were considered to represent a multituberculate (*Sigogneau-Russell, 1983a*). Three years later, the new genus and new species *Theroteinus nikolai Sigogneau-Russell, Frank & Hemmerlé, 1986* was erected and included in its monotypic family Theroteinidae *Sigogneau-Russell, Frank & Hemmerlé, 1986*. *Sigogneau-Russell, Frank & Hemmerlé (1986)* added MNHN.F.SNP 78 W, and MNHN.F.SNP 2 Ma, considered as upper teeth, in the hypodigm of the species *T. nikolai*, in association with one upper and three lower teeth, which were then not described. They studied the enamel ultrastructure and the micro-wear of these teeth and interpreted the absence of wear striations as indicating an essentially vertical masticatory movement (*Sigogneau-Russell, Frank & Hemmerlé, 1986*). *Hahn, Sigogneau-Russell & Wouters (1989)* described four new specimens (two upper and two lower teeth) and established *Theroteinus* sp. based on four lower teeth characterized by their small size, including MNHN.F.SNP 61 W along with MNHN.F.SNP 226, MNHN.F.SNP 366, and MNHN.F.SNP 497 W. All leftover specimens (n = 7) were referred to *T. nikolai* (*Hahn, Sigogneau-Russell & Wouters, 1989*).

*Hahn, Sigogneau-Russell & Wouters (1989)* included *Theroteinus*, other haramiyids, and Multituberculata *Cope (1884)* within Allotheria *Marsh (1880)*. They erected the order Theroteinida *Hahn, Sigogneau-Russell & Wouters (1989)* and raised the suborder Haramiyoidea *Hahn (1973)* to ordinal rank as Haramiyida *Hahn, Sigogneau-Russell & Wouters (1989)*. Their classification was illustrated by a phylogenetic tree in which *Theroteinus* is the sister-group of all other allotherians (*Hahn, Sigogneau-Russell & Wouters, 1989*: Text-Fig. 12). *Butler (2000)* modified the classification of Allotheria. Within the order Haramiyida of *Hahn, Sigogneau-Russell & Wouters (1989)*, he changed the rank of Theroteinida *Hahn, Sigogneau-Russell & Wouters (1989)* to suborder and put back the suborder Haramiyoidea *Hahn (1973)* (*Butler, 2000*). *Hahn & Hahn (2006)* published the most recent classification of Haramiyida including *Theroteinus*. They modified the names of the sub-orders of *Butler (2000)* as Theroteinina and Haramiyina, respectively, and included *Millsodon Butler & Hooker, 2005* (Middle Jurassic, England) into the family Theroteinidae (*Hahn & Hahn, 2006*).

In these classifications, *Theroteinus* is always considered as more basal than other haramiyids upon one main feature: in centric occlusion, one tooth of *Theroteinus* is in contact with two opposite teeth ('one-to-two' occlusion). This feature is shared by other mammaliaforms such as morganucodonts and kuehneotheriids but not by other haramiyids, which are characterized by an occlusal mode where one tooth is in contact with only one opposite tooth in centric occlusion ('one-to-one' occlusion).

## GEOLOGY AND ASSOCIATED FAUNA

The ancient sand quarry of Saint-Nicolas-de-Port, a locality close to the city of Nancy in eastern France, has yielded an abundant collection of vertebrate microremains (*Sigogneau-Russell & Hahn, 1994*). The site is part of the sandy succession of the 'Grès infraliasiques' Formation, considered as deposits in a shallow marine platform

(*Debuysschere, Gheerbrant & Allain, 2015* and references therein). The vertebrate collections of Saint-Nicolas-de-Port display a significant diversity of species belonging to Chondrichthyes, Dipnoi, Actinopterygia, Temnospondyli, Sauropsida, non-mammalian Cynodontia, and Mammaliaformes (*Debuysschere, Gheerbrant & Allain, 2015* and references therein). Saint-Nicolas-de-Port yields especially the most abundant and most diverse Upper Triassic assemblage of mammals (*Sigogneau-Russell & Hahn, 1994*; *Kielan-Jaworowska, Cifelli & Luo, 2004*; *Debuysschere, Gheerbrant & Allain, 2015*), including morganucodonts (*Debuysschere, Gheerbrant & Allain, 2015*), kuehneotheriids (*Debuysschere, 2016*), haramiyids (*Sigogneau-Russell, 1989*; *Sigogneau-Russell, 1990*), woutersiids (*Sigogneau-Russell, 1983b*; *Sigogneau-Russell & Hahn, 1995*), the problematic *Delsatia Sigogneau-Russell & Godefroit, 1997* and theroteinids that are reviewed here.

## MATERIAL

This study describes 20 isolated teeth of haramiyids from Saint-Nicolas-de-Port. Denise Sigogneau-Russell and her co-workers have excavated only one stratigraphic level in the sand pit. Specimens collected during this fieldwork are kept both in the MNHN, with the acronym 'SNP,' and in the RBINS, with the acronym 'RAS.' Several amateur palaeontologists gathered their own collections alongside Sigogneau-Russell's team and donated them to MNHN and RBINS. The collection of Georges Wouters is identified by the suffix 'W' or 'FW,' and the collection of M. Marignac is identified by the suffix 'Ma.' However, there are no data on the exact, original stratigraphic level within the quarry of these collections. All the specimens described by *Sigogneau-Russell, Frank & Hemmerlé (1986)* and *Hahn, Sigogneau-Russell & Wouters (1989)* are considered here, alongside with eight new specimens (MNHN.F.SNP 14 FW, MNHN.F.SNP 787, RBINS RAS 3 FW, RBINS RAS 11 FW, RBINS RAS 62 FW, RBINS RAS 74 FW, RBINS RAS 77 FW, RBINS RAS 103 FW).

## METHODS

### Observations, drawings and measurements

All specimens were observed with a binocular microscope (CETI, Medline Scientific, Chalgrove, United Kingdom) at a magnification power of 36. A camera lucida mounted on the microscope was used for drawings. Measurements were taken with a digital readout for metrology (Heidenhain ND 1200, Traunreut, Germany). These measurements were used to make diagrams with Excel (Microsoft, Redmond, Washington, 2013) and statistical tests with the R statistical environment (*R Development Core Team, 2016*). The 3D images of studied teeth were obtained by X-ray computed tomographic (CT) scans at the AST-RX platform of the MNHN using phoenix|x-ray|v|tome|x L 240–180 CT scanner (GE Measurement & Control Solutions, Billerica, Massachusetts) (Table S1). The 3D data were processed with Materialise Mimics Innovation Suite 17.0 Research Edition (Materialise NV, Leuven, Belgium, 2014). The SEM photos were obtained by scanning electron microscope at the RBINS using a FEI QUANTA 200 ESEM (FEI, Hillsboro, Oregon) with a voltage of 15 kV and a dwell of 10 μs.

## Dental nomenclature

The nomenclature used here to describe the haramiyid teeth is derived from *Parrington (1947*: Fig. 3*)*, *Hahn (1973*: p. 5*)*, *Butler & Macintyre (1994*: p. 435*)* and *Butler (2000*: p. 319*)*. The row of cusps named *a*/*A* is characterized by less numerous, and well-individualized cusps. This row is lingual on lower teeth, but labial on upper teeth. The second row of cusps is named *b*/B. This second row is labial on lower teeth, but lingual on upper teeth. Both rows define a central basin. Additional cusps are named *aa*/*AA* when they are on the lateral flank of row *a*/*A*, and *bb*/BB when they are on the lateral flank of row *b*/B. In each row, cusps are numbered starting from number 1. On lower teeth, the numbering starts from the mesial extremity, while on upper teeth, it starts from the distal extremity. The term 'u-ridge' refers to the junction of crests which close the basin at its distal extremity on lower teeth and its mesial extremity on upper teeth. The term 'saddle' refers to the junction of two crests which delimits the basin at its open extremity, respectively mesial on lower teeth and distal on upper teeth. This nomenclature is used only in a descriptive purpose. The homonymy does not necessarily imply homology. Capital letters are used for upper teeth and lower case letters for lower teeth.

The descriptions of the wear facets are based on the nomenclature of *von Koenigswald et al. (2013*: p. 146*)* for jaw movements. This nomenclature is used to define the direction and the angle of the slope of the wear facets. The process and the pattern of the occlusion are beyond the scope of this article and will be dealt with in detail later on.

## Methodology of characterization of Saint-Nicolas-de-Port Material

The haramiyid teeth are distinguished from other contemporary mammaliaforms by the presence of longitudinal rows of cusps separated by basins. For now, only two haramiyids are known in the Saint-Nicolas-de-Port material, *Theroteinus* and *Thomasia Poche, 1908*. The material referred to *Theroteinus* is distinguished from material referred to *Thomasia* by two main characteristics: low and obtuse cusps and a basin smaller in length and width.

All specimens described here are considered to be molariforms, by comparison with haramiyids for which premolariforms in situ are known (*Haramiyavia Jenkins et al., 1997*, *Megaconus Zhou et al., 2013*, *Arboroharamiya Zheng et al., 2013*, *Shenshou Bi et al., 2014*, and *Xianshou Bi et al., 2014*). Indeed, *Theroteinus* material does not show neither the hypertrophied mesial cusp of lower premolariform of *Haramiyavia* (*Jenkins et al., 1997*; *Luo et al., 2015*), *Megaconus* (*Zhou et al., 2013*), *Arboroharamiya* (*Zheng et al., 2013*; *Meng et al., 2014*), *Shenshou*, and *Xianshou* (*Bi et al., 2014*); nor the circular arrangement of cusps of upper premolariform of *Megaconus* (*Zhou et al., 2013*), *Arboroharamiya* (*Zheng et al., 2013*; *Meng et al., 2014*), *Shenshou*, and *Xianshou* (*Bi et al., 2014*). Moreover, *Theroteinus* teeth show no character that could give information about their position in the dental series (see below).

The orientation of *Theroteinus* teeth follows the orientation described for *Haramiyavia* (*Jenkins et al., 1997*; *Luo et al., 2015*), *Arboroharamiya* (*Zheng et al., 2013*; *Meng et al., 2014*), *Shenshou* (*Bi et al., 2014*), *Xianshou* (*Bi et al., 2014*), and *Thomasia* (*e.g.*, *Butler, 2000*). The lower molariforms are distinguished from the upper molariforms by the presence of, respectively, two or three rows of cusps and by the form of row *a*/*A*. In lower

molariforms, the lingual row *a* includes the largest cusps. Cusp *a1* is especially much larger than the other cusps, and it is located on the mesiolingual side of the crown. In upper molariforms, the labial row *A* shows three subequal cusps, when the central row *B* displays a cusp *B2* larger than the other cusps of the row and located on the distal side of the crown.

## Nomenclatural acts

The electronic version of this article in Portable Document Format (PDF) will represent a published work according to the International Commission on Zoological Nomenclature (ICZN), and hence the new names contained in the electronic version are effectively published under that Code from the electronic edition alone. This published work and the nomenclatural acts it contains have been registered in ZooBank, the online registration system for the ICZN. The ZooBank LSIDs (Life Science Identifiers) can be resolved and the associated information viewed through any standard web browser by appending the LSID to the prefix http://zoobank.org/. The LSID for this publication is: urn:lsid:zoobank.org:pub:57401966-D5B5-468C-94FD-115C0C32FE00. The online version of this work is archived and available from the following digital repositories: PeerJ, PubMed Central and CLOCKSS.

## SYSTEMATIC PALAEONTOLOGY

**Mammaliaformes** *Rowe, 1988*

Order **Haramiyida** *Hahn, Sigogneau-Russell & Wouters, 1989*

Sub-order **Theroteinida** *Hahn, Sigogneau-Russell & Wouters, 1989*

**Synonymy:** Theroteinina *Hahn & Hahn, 2006*: p. 189.

**Type-family:** Theroteinidae *Sigogneau-Russell, Frank & Hemmerlé, 1986*, by monotypy.

**Emended diagnosis:** As for the type-family.

**Distribution:** As for the type-family.

Family **Theroteinidae** *Sigogneau-Russell, Frank & Hemmerlé, 1986*

**Type-genus:** *Theroteinus Sigogneau-Russell, Frank & Hemmerlé, 1986*.

**Emended diagnosis:** As for the type-genus.

**Distribution:** As for the type-genus.

Genus ***Theroteinus*** *Sigogneau-Russell, Frank & Hemmerlé, 1986*

**Type-species:** *Theroteinus nikolai Sigogneau-Russell, Frank & Hemmerlé, 1986*.

**Referred species:** *Theroteinus rosieriensis* sp. nov.

**Emended diagnosis:** Haramiyids with lower and upper molariforms showing low cusps with more extended base and more massive aspect, short and narrow basins in relation to the size of the crown, presence of only two cusps in row *a* on lower molariforms (shared with some specimens of *Thomasia*), presence of a row *BB* on upper molariforms (potentially shared with *Eleutherodon*, *Megaconus* and *Millsodon*—see below), and an essentially vertical, masticatory movement.

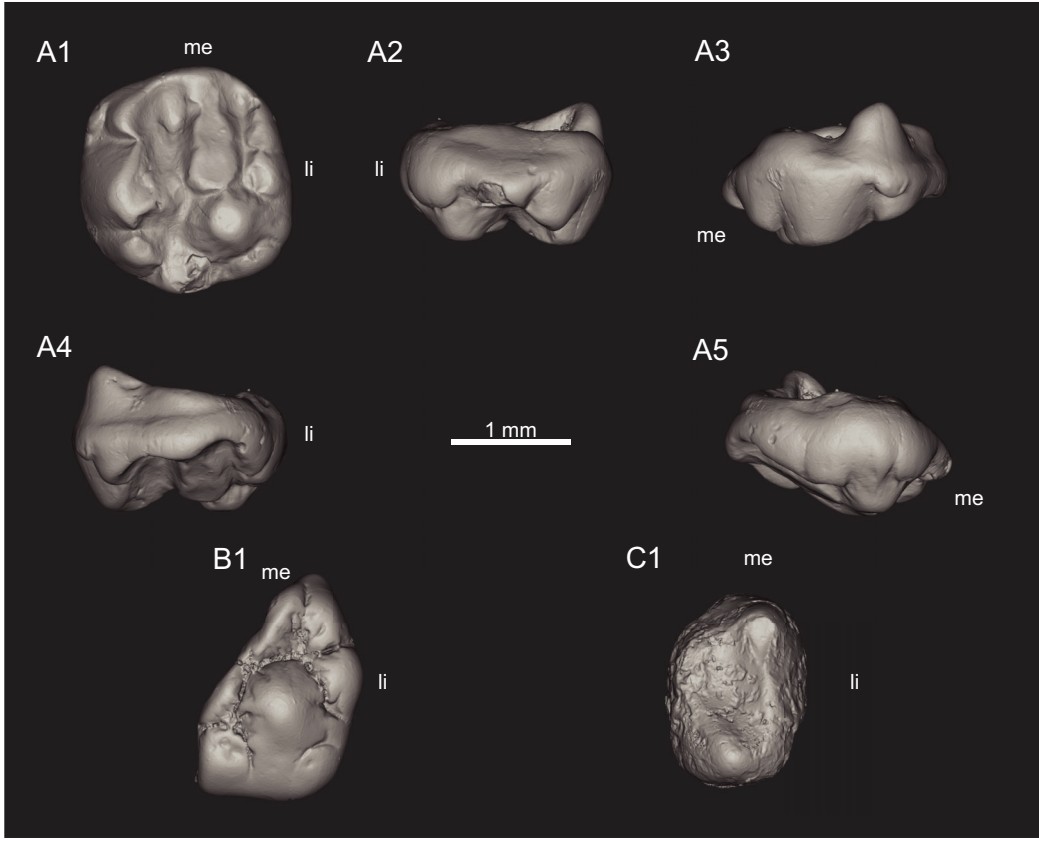

**Figure 1 Views of CT-scan reconstructions of *Theroteinus nikolai* molariforms.** (A) MNHN.F.SNP 78 W, right upper, holotype; (B) MNHN.F.SNP 722, right upper; (C) MNHN.F.SNP 226 W, left lower. 1, occlusal view; 2, distal view; 3, labial view; 4, mesial view; 5, lingual view. 'me' indicates mesial extremity and 'li' indicates lingual side.

**Distribution:** Upper Triassic (Rhaetian): France, Lorraine, Saint-Nicolas-de-Port ("Grès infraliasiques" Formation).

***Theroteinus nikolai*** *Sigogneau-Russell, Frank & Hemmerlé, 1986*

Figures 1–3

**Synonymy:** *Theroteinus* sp. *Hahn, Sigogneau-Russell & Wouters, 1989*: p. 210.

**Emended diagnosis:** *Theroteinus nikolai* differs from *T. rosieriensis* by smaller molariforms (Tables 1–3; Fig. 7A), a larger length/width ratio (Tables 1–3; Fig. 7B), a cusp *B2* more labial than the lingual basin (Figs. 1A and 3A), and a cusp *b4* more labial than the saddle (Figs. 2 and 3B–3E).

**Holotype:** MNHN.F.SNP 78 W (Figs. 1A and 3A), right upper molariform, from Saint-Nicolas-de-Port (Upper Triassic, France).

**Referred material**.

**Lower molariforms:** MNHN.F.SNP 61 W (right) (Figs. 2A and 3B), MNHN.F.SNP 226 W (left) (Fig. 1C), MNHN.F.SNP 366 W (right) (Figs. 2B and 3C), MNHN.F.SNP 497 W (right) (Figs. 2C and 3D), MNHN.F.SNP 787 (right) (Fig. 3E).
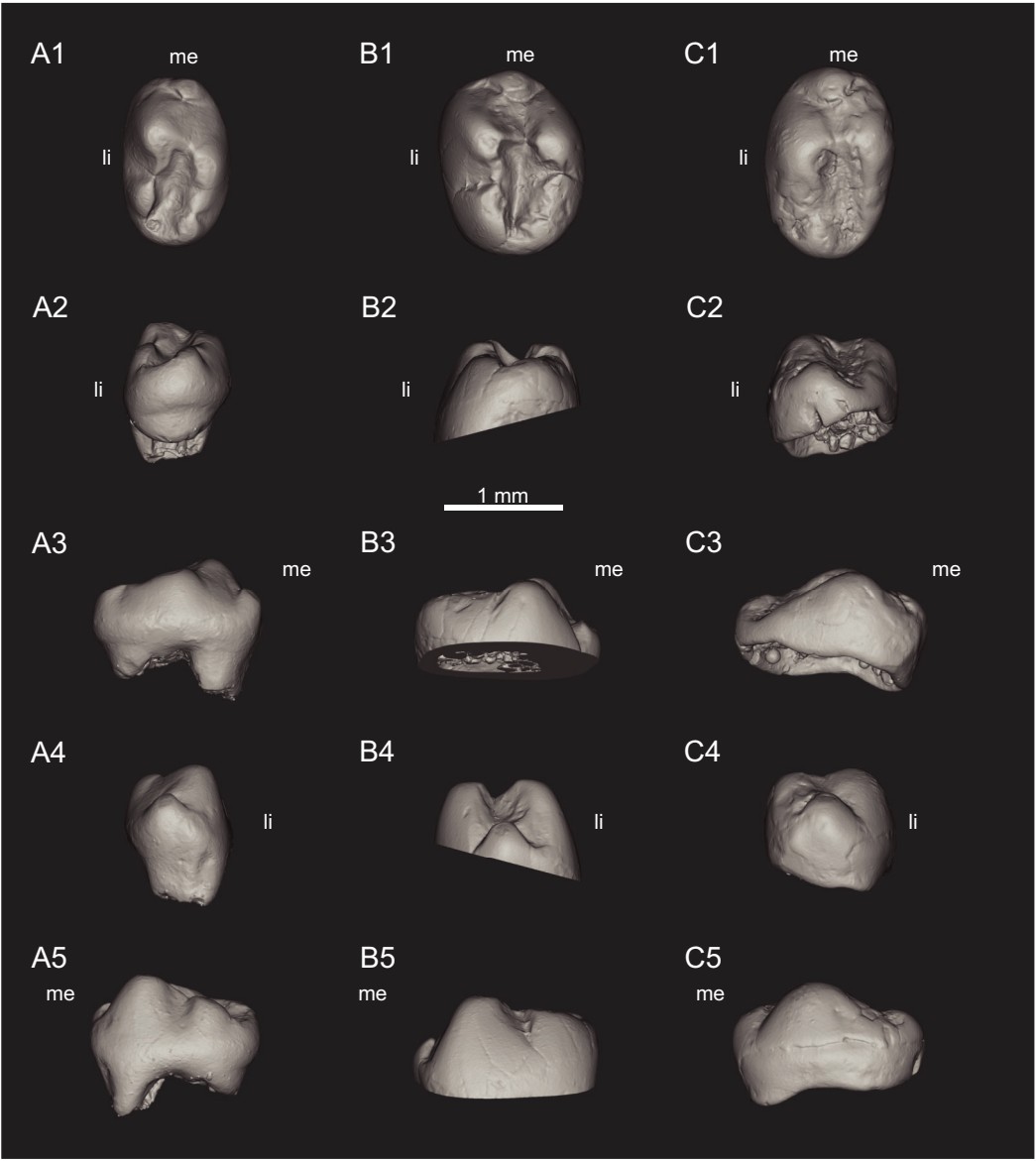

**Figure 2 Views of CT-scan reconstructions of *Theroteinus nikolai* lower molariforms.** (A) MNHN.F. SNP 61 W, right; (B) MNHN.F.SNP 366 W, right; (C) MNHN.F.SNP 497 W, right. 1, occlusal view; 2, distal view; 3, labial view; 4, mesial view; 5, lingual view. 'me' indicates mesial extremity and 'li' indicates lingual side.

**Upper molariforms:** MNHN.F.SNP 722 (right) (Fig. 1B), RBINS RAS 103 FW (right)

**Measurements:** See Table 1.

## Description

### *Lower molariforms*

The crown is dominated by two longitudinal rows of cusps which delimit a central basin, the lingual row *a* and the labial row *b*. The central basin is confined mesially by the saddle which joins cusps *a1* and *b2*, and distally by the u-ridge which joins rows *a* and *b*.

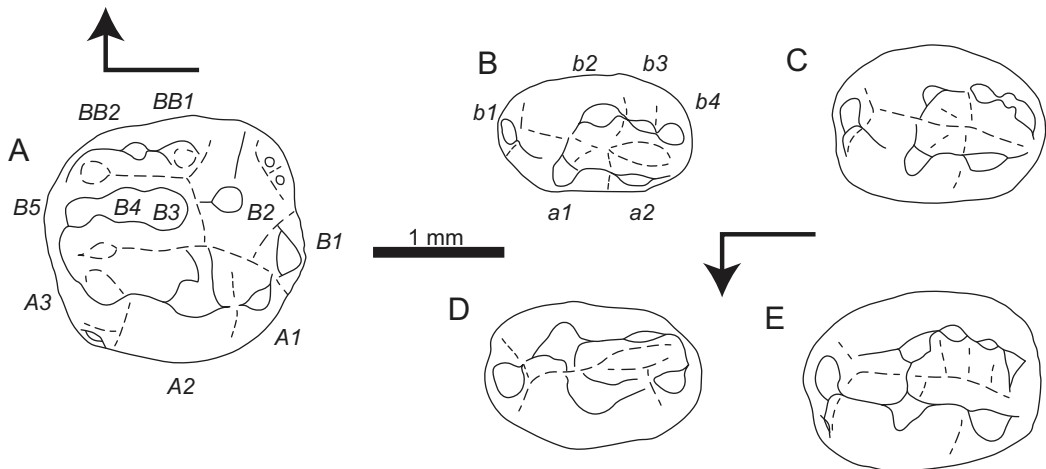

**Figure 3 Sketch drawings of *Theroteinus nikolai* molariforms in occlusal views.** (A) MNHN.F.SNP 78 W, right upper, holotype; (B) MNHN.F.SNP 61 W, right lower; (C) MNHN.F.SNP 366 W, right lower; (D) MNHN.F.SNP 497 W, right lower; (E) MNHN.F.SNP 787, right lower. Right-angled arrow indicates mesial extremity and lingual side. Letters in italics correspond to cusp nomenclature.

**Table 1 Dental measurements (in mm) of *Theroteinus* molarifoms from Saint-Nicolas-de-Port (Upper Triassic, France).**

| Material | L (mm) | W (mm) | R | Material | L (mm) | W (mm) | R |
|---|---|---|---|---|---|---|---|
| *Theroteinus nikolai* | | | | | | | |
| Upper teeth | | | | | | | |
| MNHN.F.SNP 78 W | 1.87 | 1.8 | 1.04 | RBINS.RAS 103 FW | 1.81 | 1.81 | 1.00 |
| Lower teeth | | | | | | | |
| MNHN.F.SNP 61 W | 1.46 | 0.94 | 1.54 | MNH.F.SNP 226 W | 1.88 | 1.23 | 1.53 |
| MNHN.F.SNP 366 W | 1.57 | 1.3 | 1.21 | MNHN.F.SNP 497 W | 1.65 | 1.15 | 1.44 |
| MNHN.F.SNP 787 | 1.88 | 1.36 | 1.38 | | | | |
| *Theroteinus rosieri* | | | | | | | |
| Upper teeth | | | | | | | |
| MNHN.SNP 2 Ma | 2.41 | | | MNHN.SNP 335 W | 2.18 | 2.49 | 0.88 |
| RBINS.RAS 801 | 2.23 | 2.43 | 0.91 | | | | |
| Lower teeth | | | | | | | |
| MNHN.SNP 309 W | 2.05 | 1.98 | 1.03 | MNHN.F.SNP 487 W | 2.53 | 2.15 | 1.18 |
| RBINS.RAS 3 FW | | 1.92 | | RBINS.RAS 11 FW | 2.21 | 1.79 | 1.23 |
| RBINS.RAS 62 FW | 1.87 | 1.38 | 1.36 | RBINS.RAS 74 FW | 2.11 | 1.64 | 1.28 |
| RBINS.RAS 77 FW | 2.02 | 1.67 | 1.21 | RBINS.RAS 800 | 2.25 | 2.00 | 1.13 |

**Note:**
L, mesiodistal length; W, labiolingual width; R, length/width ratio.

The central basin gets deeper and narrower from the mesial extremity to the distal extremity.

Cusp row *a* includes two cusps. Cusp *a1* is the largest one of the teeth and rises vertically in lateral view. This cusp extends over the mesial half of the tooth. Cusp *a1* shows a weak mesial carina which splits into two segments. One segment runs mesially and the other bends labially to join cusp *b1*. At the level of the base of cusp *b1*, the mesial segment turns

**Table 2 Means, standard deviations and medians for length, width (in mm), and length/width ratio for molariforms of *Theroteinus* from Saint-Nicolas-de-Port (Upper Triassic, France).**

| Taxa | Series | Measurements | Means | Standard deviations | Medians |
|------|--------|--------------|-------|---------------------|---------|
| *T. nikolai* | Upper | Length (mm) | 1.8401 | 0.0407 | 1.8401 |
| | | Width (mm) | 1.8045 | 0.0097 | 1.8045 |
| | | Length/width | 1.0198 | 0.028 | 1.0198 |
| | Lower | Length (mm) | 1.6872 | 0.1871 | 1.6484 |
| | | Width (mm) | 1.1953 | 0.1619 | 1.2256 |
| | | Length/width | 1.4211 | 0.136 | 1.4389 |
| *T. rosieriensis* | Upper | Length (mm) | 2.2736 | 0.1211 | 2.2264 |
| | | Width (mm) | 2.4628 | 0.0407 | 2.4628 |
| | | Length/width | 0.8955 | 0.0272 | 0.8955 |
| | Lower | Length (mm) | 2.1482 | 0.2101 | 2.1098 |
| | | Width (mm) | 1.8179 | 0.2472 | 1.8585 |
| | | Length/width | 1.2023 | 0.1054 | 1.2098 |

**Table 3 Statistical comparisons of the means of lower molariforms of *Theroteinus nikolai* and *Theroteinus rosieriensis* from Saint-Nicolas-de-Port (Upper Triassic, France) by *t*-test.** Normality of the data has been tested by Shapiro-Wilk Test (Table S2), p-values have been corrected by Holm-Bonferroni method, the alternative hypothesis is "true difference in means is not equal to 0."

| Measurements tested | Value of the test ($t$) | 95% confidence interval | p-value | Adjusted p-value |
|---------------------|-------------------------|-------------------------|---------|------------------|
| Length | −3.9959 | −0.7204336; −0.2016236 | 0.002882 | 0.005764* |
| Width | −5.4858 | −0.8726922; −0.3725728 | 0.0001959 | 0.0005877* |
| Length/width | 3.0109 | 0.04829294; 0.38939849 | 0.01875 | 0.01875* |

**Note:**
* Indicates statistically significant results (threshold = 0.05).

into a short, horizontal cingulum to join cusp *b1*. A distal crest starts from the distolabial side of the apex of cusp *a1* to join cusp *a2*. This crest is straight in lateral view, but it is curved labially in occlusal view. A sulcus underlines the lingual side of this crest and descends to the base of cusp *a1*. A second crest, straight in occlusal and lateral views, starts from the labial side of the apex of cusp *a1* to the base, where it takes part in the saddle. The distal and labial crests delimit a concave, narrow surface on the distolabial flank of cusp *a1*, which extends from the apex to the central basin. Cusp *a2* is twice lower and labiolingually narrower, and much mesiodistally shorter than cusp *a1*. Cusp *a2* is more lingual than cusp *a1*. The lingual flanks of cusps *a1* and *a2* are aligned and parallel to the mesiodistal axis of the tooth on MNHN.F.SNP 61 W, but deviate distolabially on MNHN.F.SNP 366 W, and MNHN.F.SNP 787. The occlusal outline of cusp *a2* is semicircular with a convex, lingual side and nearly flat, labial side. The labial side shows a vertical, weak ridge in the middle. In distal view, the slope of the labial flank is more vertical than the slope of the lingual flank. The latter is slightly convex. In labial view, the mesial base of cusp *a2* is higher than the distal base of the cusp. In lingual view, the bases of cusps *a1* and *a2* are at the same level. Cusp *a2* shows two crests, respectively mesial and distal, straight in lateral and occlusal views, and aligned mesiodistally. The first crest starts from

the mesiolabial side of the apex to join the distolingual crest of cusp *a1*. The second crest starts from the distolabial side of the apex to the extremity of row *a*. The distal crest is much longer than the mesial crest. The slope of the mesial crest of cusp *a2* is weaker than the slope of the distal crest of cusp *a1* and the slope of the distal crest of cusp *a2*. The slope of the latter is more vertical than the slope of distal crest of cusp *a1*.

Cusp row *b* includes four cusps, less distinguished from each other than the cusps of row *a*. Cusp *b1* is the most mesial of the tooth. This cusp is subequal in size with cusp *b4*, or larger in MNHN.F.SNP 61 W. Cusp *b1* is located in front of the saddle, but tends to rise lingually to join the mesiolabial carina of cusp *a1*. Cusp *b2* is the largest cusps of row *b*. This cusp is slightly smaller than cusp *a1*, except in MNHN.F.SNP 366 W where cusp *b2* is slightly larger and higher than cusp *a1*. Cusp *b2* is labial to cusp *a1*, its base extends as mesially but much less distally, and its apex is slightly more distal, or much more distal in MNHN.F.SNP 61 W. Cusp *b2* shows two crests, straight in occlusal and lateral views. The first crest runs labially, but mesiolabially in MNHN.F.SNP 61 W, to take part in the saddle. The second crest runs distally and joins cusp *b3*. Both crests define on the one side a slightly convex, distolingual occlusal outline, and on the other side a large arc of a circle. Cusp *b3* is much smaller than cusps *a1*, *a2*, and *b2* and slightly smaller than cusps *b1* and *b4*. Cusp *b3* is directly distal to cusp *b2*, except in MNHN.F.SNP 787 where it is slightly more labial. This cusp is more mesial than *a1–a2* notch in MNHN.F.SNP 787, aligned with *a1–a2* notch in MNHN.F.SNP 366 W, and more distal than *a1–a2* notch in MNHN.F.SNP 61 W. The apex of cusp *b3* is slightly higher than the apex of cusp *a2*, or at the same level in MNHN.F.SNP 61 W. The long axis of cusp *b3* slightly deviates distolingually from the mesiodistal axis of the tooth, except in MNHN.F.SNP 787 where both axes are parallel. Cusp *b4* is distal to cusp *b3* and slightly more lingual. Consequently, the long axes of both cusps are aligned, except in MNHN.F.SNP 787. The apex of cusp *b4* takes place slightly lower than the apex of cusp *a2* and faces the distal crest of cusp *a2*. Cusp *b4* shows a lingual carina, which is well developed in MNHN.F.SNP 787. The u-ridge is a low crest which extends row *b* and bends lingually to join the extremity of row *a*.

*Comments on MNHN.F.SNP 266 W and MNHN.F.SNP 497 W*

MNHN.F.SNP 266 W and MNHN.F.SNP 497 W are difficult to describe because they are extensively damaged. The surface of MNHN.F.SNP 266 W is not well preserved (Fig. 1C) and MNHN.F.SNP 497 W is heavily worn. For these reasons, they have not been incorporated in the description above. About MNHN.F.SNP 497 W, it may be noticed that row *b* is less developed than in other specimens, with a strong reduction of cusp *b4* (Figs. 2C and 3D). In the absence of clear morphological characters, both specimens are referred to *Theroteinus nikolai* following morphometry (see Comparisons. Identification of *Theroteinus* species below).

### Upper molariforms

The crown is dominated by three longitudinal rows of cusps: labial row *A*, central row *B* and lingual row *BB*. Rows *A* and *B* define a labial basin delimited distally by the saddle,

constituted only by the lingual crest of cusp *A2*, and mesially by the u-ridge which joins rows *A* and *B*. Rows *B* and *BB* define a lingual basin, smaller than the labial basin, delimited distally by the meeting of cusps *B2* and *BB1*, and mesially by the crest which joins rows *B* and *BB*. Both basins get deeper and larger mesially.

Row *A* includes three mesiodistally aligned cusps. The three cusps take place at the same level on the crown. Cusps *A1* and *A3* are subequal in length and width, cusp *A1* is slightly higher than cusp *A3*. Cusp *A2* is twice mesiodistally longer and higher, and labiolingually much wider than cusps *A1* and *A3*. The occlusal outlines of cusps *A1* and *A3* show a semicircular, labial flank and a relatively flat, lingual flank, sometimes concave because of wear. Cusp *A1* shows two crests, straight in occlusal and lateral views. The longest crest runs distolingually from the apex to cusp *B1*. The other crest runs mesially to cusp *A2*. The slopes of these crests are subequal. Cusp *A3* shows two crests, straight in occlusal and lateral views. The longest crest runs mesiolingually from the apex to take part in the u-ridge. The other crest runs distally to cusp *A2*. The slopes of these crests are subequal. Cusp *A2* shows three crests, straight in occlusal and lateral views. The first crest runs distally to cusp *A1*. The second crest runs mesially to cusp *A3*. The third crest runs lingually but does not join another structure. The distal crest is the shortest and shows the strongest slope. The lingual crest is much wider than both other crests. The lingual and mesial crests define a flat surface on the mesiolingual flank of cusp *A2*. *A1*–*A2* and *A2*–*A3* notches are equal in depth, but *A1*–*A2* notch takes place higher than *A2*–*A3* notch.

A small, supplementary cusp takes place under the labial flank of cusp *A3*.

Row *B* includes five cusps. Cusp *B1* looks like a distally curved semicircle. This cusp is slightly smaller in length and width than cusp *A1*, but much smaller in height. Cusp *B1* is more distal and more lingual than cusp *A1* and is mesiodistally aligned with the saddle. Cusp *B2* is slightly smaller than cusp *A2*. Cusp *B2* is much more lingual than cusp *B1* and is labiolingually aligned with the *A1*–*A2* notch. This cusp is cone-shaped and does not show any crest. Three small cuspules take place at the base of the mesiolingual flank of cusp *B2*. Cusp *B3* is directly mesial to cusp *B2*. The apex of cusp *B3* is slightly more labial the apex of cusp *B2* and slightly more mesial than apex of cusp *A2*. Cusp *B3* is subequal in size with cusp *B1* and take place slightly lower than cusp *B2*. Cusp *B4* is directly mesial to cusp *B3*. This cusp is smaller in all dimensions and takes place lower than cusp *B3*. Cusp *B4* is labiolingually aligned with *A2*–*A3* notch. Cusp *B5* is the most mesial cusp of the tooth. This cusp is smaller in all dimensions, takes place lower, and is slightly more labial than cusp *B4*. The mesial extremity of cusp *B5* shows two crests. One crest runs labially to take part in the u-ridge of the labial basin. The other crest runs lingually to mesially close the lingual basin.

Row *BB* includes two cusps. Cusp *BB1* is sub-equal in size to cusp *B4* and takes place at the same level. Cusp *BB1* is placed right next to cusps *B2* and *B3*, directly lingual to the *B2*–*B3* notch. Cusp *BB2* is mesial to cusp *BB1* but slightly more lingual. This cusp is subequal in length and width with cusp *B5,* but slightly higher. A cusp *BB3* was possibly present, but this part of the crown is broken. Row *BB* is extended by a crest which runs mesially and then bends labially to close the lingual basin.

*Comments on MNHN.F.SNP 722*

Only the distal part of MNHN.F.SNP 722 is preserved, with cusps *A1*, *B1*, *B2*, *BB1*, and a part of cusps *A2* and *B3* (Fig. 1B). Since morphometry is not applicable, this specimen is referred to *Theroteinus nikolai* following the position of cusp *B2* in relation to cusps *B3* and *BB1*. However, some doubts remain because cusp *B3* is fragmentary. MNHN.F.SNP 722 differs from MNHN.F.SNP 78 W by a smaller cusp *A1* and presence of only one cuspule at the base of the mesiolingual flank of cusp *B2*.

## Wear

### Lower molariforms

In MNHN.F.SNP 787, only the apices of the cusps are abraded by wear. In MNHN.F.SNP 366 W, the apices of cusps *a1* and *a2* show a shallow, distal facet. The apex of cusp *b1* shows a steep, mesiolabial facet. The apex of cusp *b2* shows a shallow, distal facet. Cusp *b3* shows a steep, labial facet, which slightly extends on the mesial part of cusp *b4*. Cusp *b4* shows a shallow, distal facet. The sides of the basin show traces of wear but do not develop clear facets. In MNHN.F.SNP 61 W, the facets of apices of the cusps *a1* and *b2* are more extended labially and the carina of cusp *a1* is flattened. MNHN.F.SNP 61 W differs from MNHN.F.SNP 366 W by a horizontal facet on cusp *b3*. In MNHN.F.SNP 497 W, the distal parts of cusps *a1* and *a2* each show a shallow, distal facet. The distal part of cusps *b2*, *b3*, and *b4* shows one shallow, distal facet. The remains of cusps *a1* and *b2*, and cusp *b1* are abraded by wear.

### Upper molariforms

In MNHN.F.SNP 722, only the apices of the cusps are abraded by wear. MNHN.F.SNP 78 W shows a large number of well-defined facets. The apex of cusp *A1* shows a steep, distolingual facet. The apex of cusp *A2* shows a horizontal, mesial facet. This facet is connected with traces of wear on the mesiolingual side of the cusp which spread from the apex to the labial basin. The lingual crest of cusp *A2* is slightly flattened by wear. The lingual side of cusp *A3* is truncated by a concave, steep, lingual facet. Cusp *B1* shows diffuse traces of wear but no distinct wear facet. The apex of cusp *B2* shows a horizontal, mesial facet. The mesiolingual and labial sides of cusp *B2* show slight traces of wear. The apex of cusp *B3* shows a shallow, mesio-mesiolingual facet. The apex of cusp *B4* shows a steep, mesio-mesiolingual facet. Cusp *B5* shows a concave, shallow, lingual facet. The apex of cusp *BB1* possibly shows a horizontal facet but is partially broken. Cusp *BB2* shows a steep, mesial facet. The flanks of the labial basin show traces of wear.

***Theroteinus rosieriensis*** sp. nov. urn:lsid:zoobank.org:act:F3C6B3B3-1733-4625-942F-9C085A51116A

Figures 4–6

**Etymology:** rosieri–: a Latinized form of 'Rosières' from 'Rosières-aux-Salines' another name used for the study site; -ensis; suffix added to a toponym to form an adjective.

**Diagnosis:** *Theroteinus rosieriensis* differs from *T. nikolai* by larger molariforms (Tables 1–3; Fig. 7A), a lower length/width ratio (Tables 1–3; Fig. 7B), a cusp *B2* mesiodistally aligned

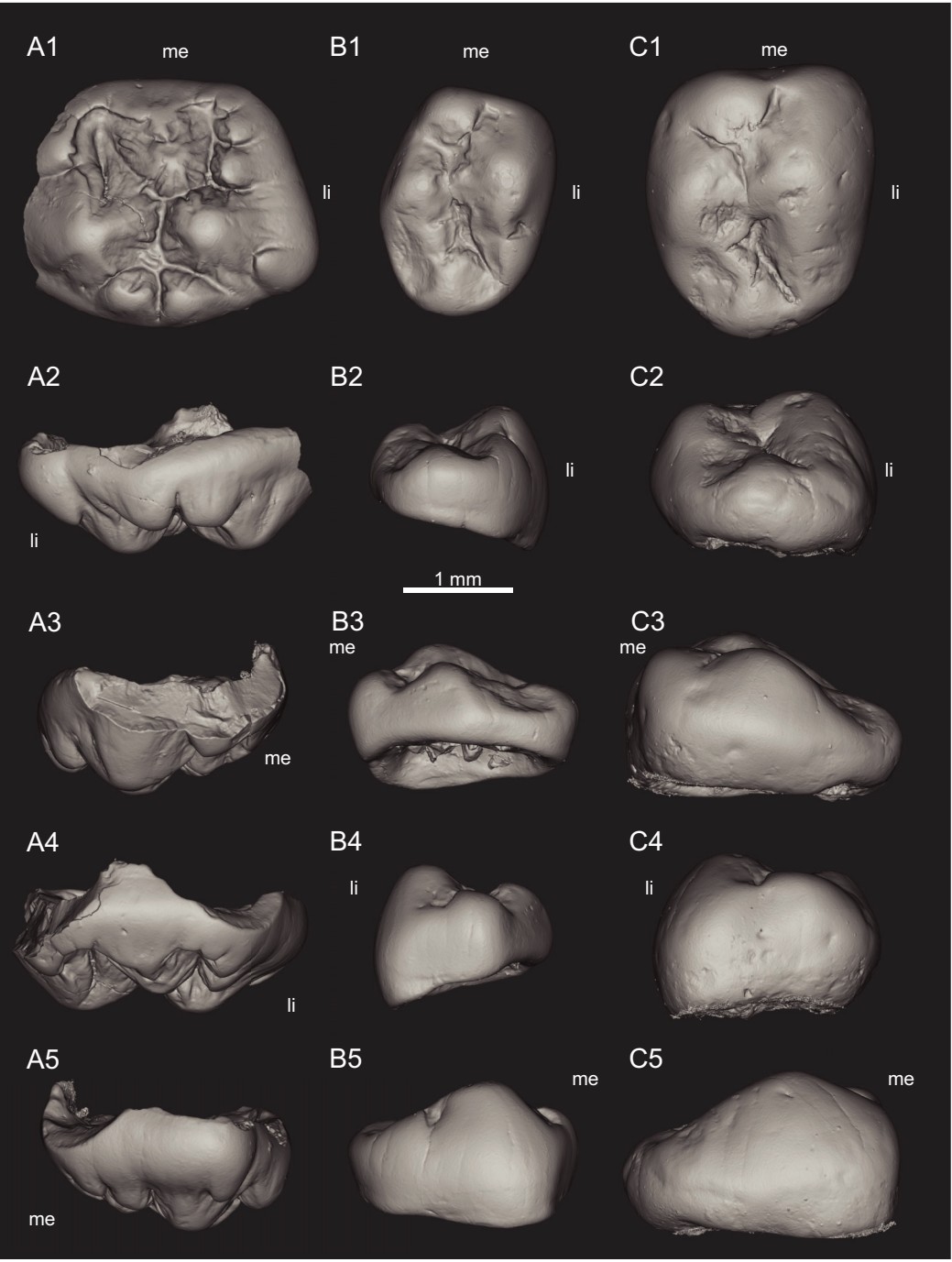

**Figure 4 Views of CT-scan reconstructions of *Theroteinus rosieriensis* molariforms.** (A) MNHN.F. SNP 2 Ma, right upper, holotype; (B) MNHN.F.SNP 309 W, left lower; (C) MNHN.F.SNP 487 W, left lower. 1, occlusal view; 2, distal view; 3, labial view; 4, mesial view; 5, lingual view. 'me' indicates mesial extremity and 'li' indicates lingual side.

with the lingual basin (Figs. 1A and 3A), and a cusp *b4* mesiodistally aligned with the saddle (Figs. 2 and 3B–3E).

**Holotype:** MNHN.F.SNP 2 Ma (Figs. 4A and 6A), right upper molariform.

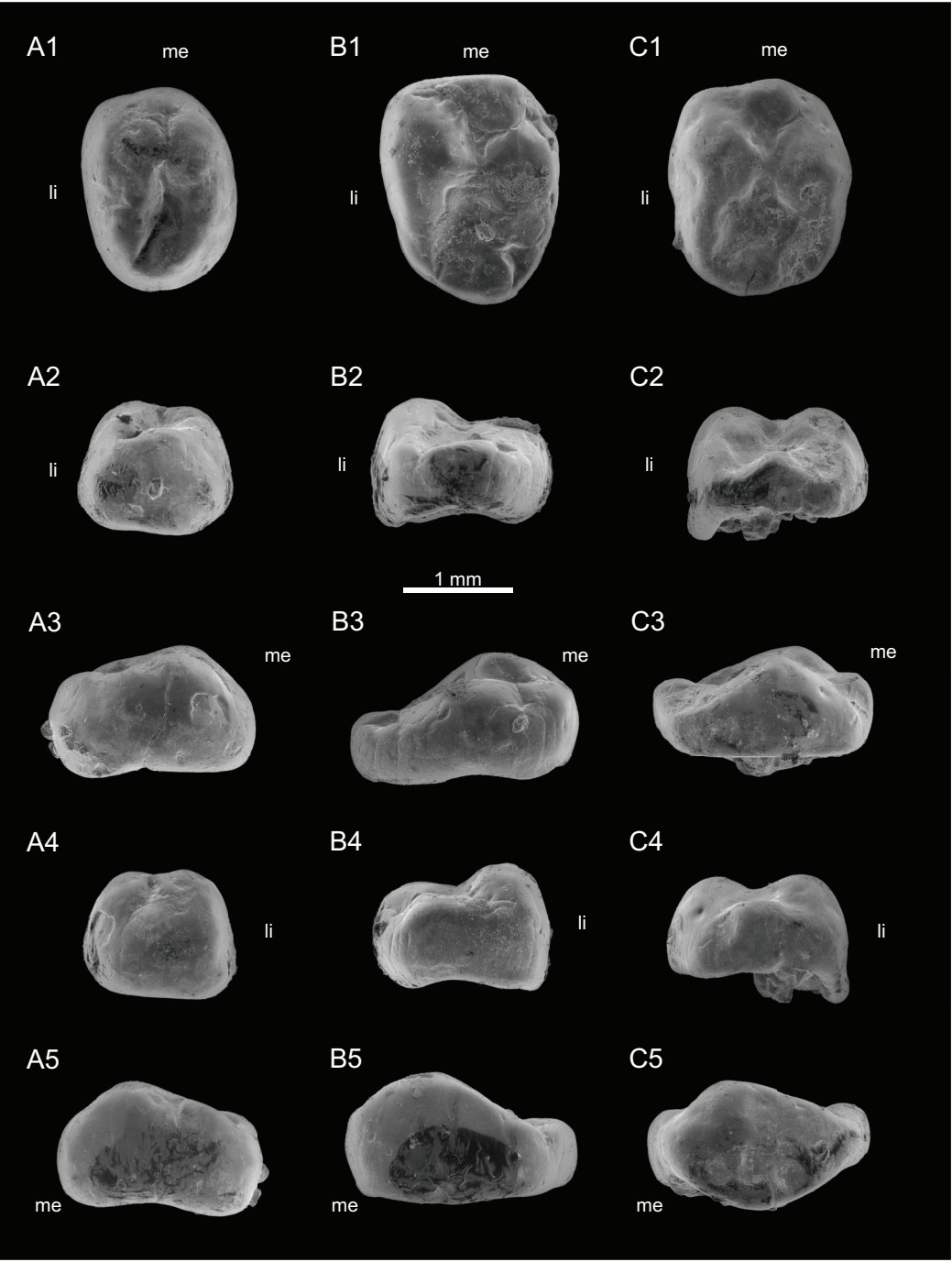

**Figure 5 SEM photographs of *Theroteinus rosieriensis* lower molariforms.** (A) RBINS.RAS 62 FW, right; (B) RBINS.RAS 74 FW, right; (C) RBINS.RAS 77 FW, right. 1, occlusal view; 2, distal view; 3, labial view; 4, mesial view; 5, lingual view. 'me' indicates mesial extremity and 'li' indicates lingual side.

**Type locality and horizon:** Saint-Nicolas-de-Port Quarry, around 1 km south-southeast of Saint-Nicolas-de-Port city, Meurthe-et-Moselle department, Lorraine Region, France. Sands of 'Grès infraliasiques' Formation, Rhaetian, Upper Triassic.

**Referred material.**

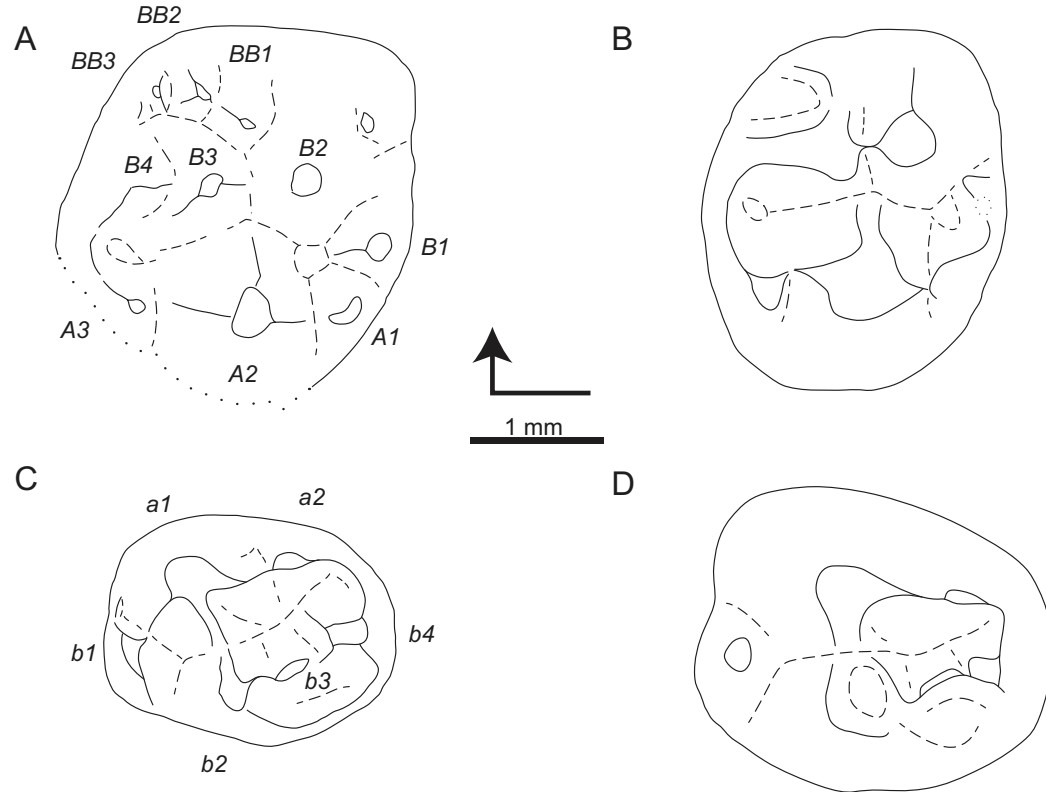

**Figure 6 Sketch drawings of *Theroteinus rosieriensis* molariforms in occlusal views.** (A) MNHN.F. SNP 2 Ma, right upper, holotype; (B) MNHN.F.SNP 335 W, right upper; (C) MNHN.F.SNP 309 W, left lower; (D) MNHN.F.SNP 487 W, left lower. Right-angled arrow indicates mesial extremity and lingual side. Letters in italics correspond to cusp nomenclature.

**Lower molariforms:** MNHN.F.SNP 309 W (left) (Figs. 4B and 6C), MNHN.F.SNP 487 W (left) (Figs. 4C and 6D), RBINS.RAS 3 FW (right), RBINS.RAS 11 FW (left), RBINS.RAS 62 FW (right) (Fig. 5A), RBINS.RAS 74 FW (right) (Fig. 5B), RBINS 77 FW (right) (Fig. 5C), RBINS.RAS 800 (right).

**Upper molariforms:** MNHN.F.SNP 14 FW (right), MNHN.F.SNP 335 W (right) (Fig. 6B), RBINS.RAS 801 (left).

**Measurements:** See Table 1.

## Description

### Lower molariforms

The crown is dominated by two longitudinal rows of cusps which delimit a central basin, the lingual row *a* and the labial row *b*. The central basin is confined mesially by the saddle which joins cusps *a1* and *b2* and distally by the u-ridge which joins rows *a* and *b*. The saddle is very high compared with the u-ridge, except in MNHN.F.SNP 309 W where the difference is weaker. The central basin gets deeper and narrower from the mesial extremity to the distal extremity.

Row *a* includes two cusps. Cusp *a1* is the largest cusp of the tooth and rises vertically in lateral view. This cusp extends over the mesial half of the tooth, even more in MNHN.F.

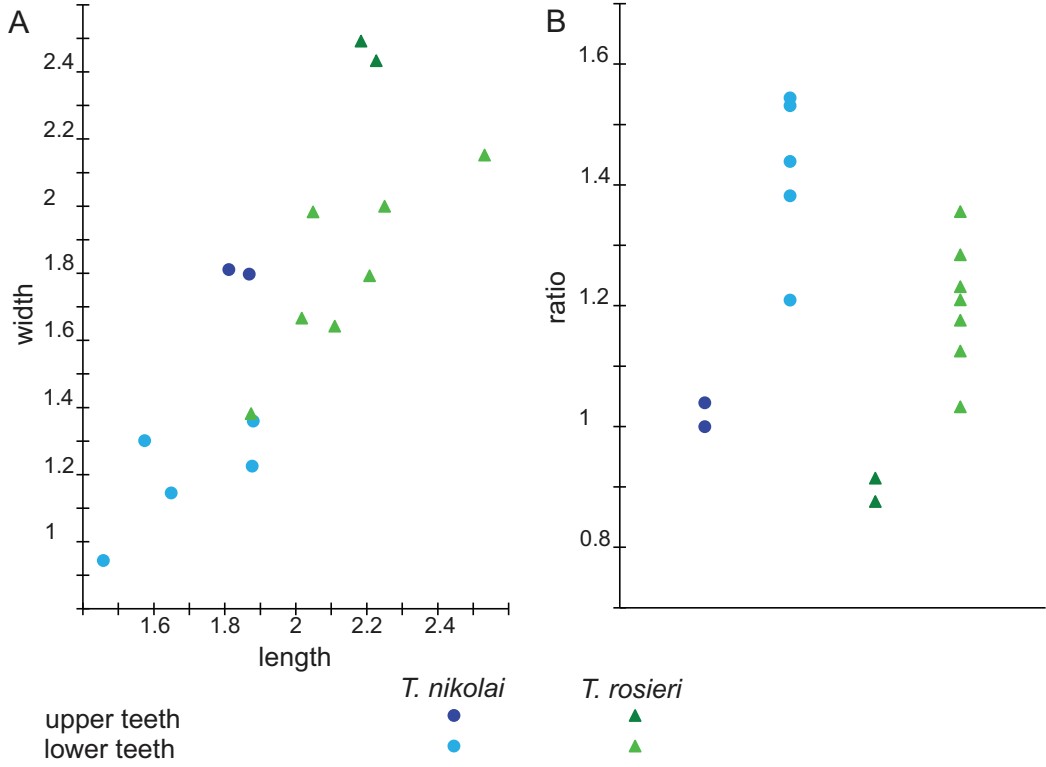

**Figure 7** Scatterplots of Theroteinus specimens from Saint-Nicolas-de-Port according to (A) length, width (in mm) and (B) length/width ratio (measurements in Table 1).

SNP 487 W, RBINS.RAS 74 FW, and RBINS.RAS 77 FW. Cusp *a1* shows a weak mesial carina which splits into two segments. One segment runs mesially and the other bends labially to join Cusp *b1*. At the level of the base of cusp *b1*, the mesial segment turns into a short, horizontal cingulum to join cusp *b1*. A distal crest starts from the distolabial side of the apex of cusp *a1* to join cusp *a2*. This crest is straight in lateral view, but it is curved labially in occlusal view, except in MNHN.F.SNP 487 W, RBINS.RAS 62 FW, and RBINS. RAS 74 FW where it is straight in both views. A sulcus underlines the lingual side of this crest and descends to the base of cusp *a1*, absent in MNHN.F.SNP 487 W and RBINS. RAS 74 FW. A second crest, straight in occlusal and lateral views, starts from the labial side of the apex of cusp *a1* to the base, where it takes part in the saddle. The distal and labial crests delimit a concave, narrow surface on the distolabial flank of cusp *a1*, which extends from the apex to the central basin. Cusp *a2* is half as cusp *a1* in height, labiolingual width, and mesiodistal length, even less in RBINS.RAS 74 FW. Cusp *a2* is more lingual than cusp *a1*. The lingual flanks of cusps *a1* and *a2* are aligned and deviate distolabially from the mesiodistal axis of the tooth. The occlusal outline of cusp *a2* is semicircular with a convex, lingual side and nearly flat, labial side, except in MNHN.F. SNP 487 W because of wear. In distal view, the slopes of the labial and lingual flanks are subequal. The latter is slightly convex. In labial view, the mesial base of cusp *a2* is higher than the distal base of the cusp. In lingual view, the bases of cusps *a1* and *a2* are at the same level. Cusp *a2* shows two crests, respectively mesial and distal, straight in lateral

and occlusal views, and aligned mesiodistally. The first crest starts from the mesiolabial side of the apex to join the distolingual crest of cusp *a1*. The second crest starts from the distolabial side of the apex to the extremity of row *a*. The distal crest is much longer than the mesial crest. The slope of the mesial crest of cusp *a2* is weaker than the slope of the distal crest of cusp *a1* and the slope of the distal crest of cusp *a2*. The slope of the latter is more vertical than the slope of the distal crest of cusp *a1*. These crests are not preserved in MNHN.F.SNP 487 W and RBINS.RAS 74 FW.

Row *b* includes four cusps, less distinguished from each other than the cusps of row *a*. Cusp *b1* is the most mesial of the tooth. This cusp is subequal in high and mesiodistal length with cusp *a2* but wider and more voluminous. Cusp *b1* is located in front of the saddle, but tends to rise lingually to join the mesiolabial carina of cusp *a1*. Cusp *b2* is the largest cusps of row *b*. This cusp is slightly smaller than cusp *a1*, except in MNHN.F. SNP 309 W and RBINS.RAS 62 FW where it is much smaller but still larger than other cusps. Cusp *b2* is labial to cusp *a1*, its base extends less mesially and distally, and its apex is slightly more distal, or much more distal in MNHN.F.SNP 309 W. Cusp *b2* shows two crests, straight in occlusal and lateral views. The first crest runs labially to take part in the saddle. The second crest runs distally and joins cusp *b3*. Both crests define on the one side a slightly convex, distolingual occlusal outline, and on the other side a large arc of a circle. This part of the crown has been removed by wear in MNHN.F.SNP 487 W. Cusp *b3* is much smaller than cusps *a1*, *a2*, *b1*, and *b2* and slightly smaller than cusp *b4*. Cusp *b3* is distal and slightly lingual to cusp *b2*. This cusp is labiolingually aligned with the *a1–a2* notch. The base of cusp *b3* is slightly lower than the base of cusp *a2*. The long axis of cusp *b3* deviates slightly distolingually from the mesiodistal axis of the tooth. This part of the crown has been removed by wear in RBINS.RAS 74 FW and RBINS.RAS 77 FW. Cusp *b4* is distal to cusp *b3* and slightly more lingual. Cusp *b4* is mesiodistally aligned with the saddle and labiolingually aligned with the distal crest of cusp *a2*. The base of cusp *b4* is slightly lower than the base of cusp *b3*. In MNHN.F.SNP 487 W, RBINS.RAS 62 FW, and RBINS.RAS 77 FW, a low crest extends row *b* and bends lingually to join the extremity of row *a*. In MNHN.F.SNP 309 W and RBINS.RAS 74 FW, this crest splits into two segments. The first segment bends lingually to join the extremity of row *a*. The second segment bends labially and runs down the side of the crown and turns into a thin bulge which extends into the base of cusp *b2*.

*Comments on RBINS.RAS 800*

The occlusal surface of RBINS.RAS 800 is not well preserved. As a consequence, the cusps are difficult to describe. For these reasons, this specimen has not been included in the description above. In the absence of clear morphological characters, this specimen is referred to *Theroteinus rosieriensis* following morphometry (see Comparisons. Identification of *Theroteinus* species below).

**Upper molariforms**

The crown is dominated by three longitudinal rows of cusps: labial row *A*, central row *B* and lingual row *BB*. Rows *A* and *B* define a labial basin delimited distally by the saddle,

constituted only by the lingual crest of cusp *A2*, and mesially by the u-ridge which joins rows *A* and *B*. Rows *B* and *BB* define a lingual basin, smaller than the labial basin, delimited distally by the meeting of cusps *B2* and *BB1*, and mesially by the crest which joins rows *B* and *BB*. Both basins get deeper and larger mesially. The lingual basin is very shallow in MNHN.F.SNP 335 W.

Row *A* includes three cusps. In MNHN.F.SNP 335 W, cusp *A3* is slightly more labial than cusps *A1* and *A2*. In MNHN.F.SNP 2 Ma, cusp *A1* is more lingual than cusp *A2* and cusp *A3* is more labial than cusp *A2*. The three cusps are located at the same level on the crown. Cusps *A1* and *A3* are subequal in height and width, cusps *A1* is slightly longer than cusp *A3*. Cusp *A2* is twice mesiodistally longer and higher, and much labiolingually wider than cusps *A1* and *A3*. In MNHN.F.SNP 2 Ma, cusps *A1* and *A3* are less wide than cusp *A2*. In occlusal view, cusps *A1* and *A3* show a semicircular labial flank and a relatively flat, lingual flank. Cusp *A1* shows two crests, straight in occlusal and lateral views. The longest crest runs distolingually from the apex to cusp *B1*. The other crest runs mesially to cusp *A2*. In MNHN.F.SNP 2 Ma, the mesial crest is present but cusp *A1* shows a flat side in front of cusp *B1*. Cusp *A3* shows two crests, straight in occlusal and lateral views. The longest crest runs mesiolingually from the apex to take part in the u-ridge. The other crest runs distally to cusp *A2*. The slope of the distal crest is more vertical than the slope of the mesial crest. Cusp *A2* shows three crests, straight in occlusal and lateral views. The first crest runs distally to cusp *A1*. The second crest runs mesially to cusp *A3*. The third crest runs lingually but does not join another structure. The distal crest is the shortest, and the slopes of the three crests are subequal. The lingual crest is much wider than both other crests. The lingual and mesial crests define a concave surface on the mesiolingual flank of cusp *A2*. *A1–A2* notch is less deep and takes place higher than *A2–A3* notch.

Row *B* includes four cusps. Cusp *B1* is subequal in size with cusp *A1* in MNHN.F.SNP 2 Ma, but smaller in MNHN.F.SNP 335 W. Cusp *B1* is more distal and more lingual than cusp *A1* and is mesiodistally aligned with the saddle. Cusp *B2* is slightly smaller than cusp *A2*. Cusp *B2* is much more lingual than cusp *B1* and is labiolingually aligned with the *A1–A2* notch. This cusp is cone-shaped and does not show any crest. One small cuspule takes place at the base of the mesiolingual flank of cusp *B2*. *B2–B3* notch is labiolingually aligned with cusp *A2*. Cusp *B3* is more labial than cusp *B2* and slightly more lingual than cusp *B1* (MNHN.F.SNP 2 Ma) or mesiodistally aligned to cusp *B1* (MNHN.F. SNP 335 W). Cusp *B3* is much smaller than cusps *A2* and *B2* and slightly larger than cusps *A1*, *A3*, and *B1* (MNHN.F.SNP 2 Ma) or subequal with cusps *A1* and *A3* (MNHN.F.SNP 335 W). In MNHN.F.SNP 2 Ma, cusp *B3* is wider than long. Cusp *B3* takes place slightly lower than cusp *B2*. *B3–B4* notch is labiolingually aligned with the *A2–A3* notch. Cusp *B4* is directly mesial to cusp *B3*. This cusp is smaller in all dimensions and located lower than cusp *B3*. In MNHN.F.SNP 335 W, a cusp *B5* was potentially present but is now removed by wear. The mesial extremity of row *B* shows two crests. One crest runs labially to take part in the u-ridge of the labial basin. The other crest runs lingually and closes the lingual basin at the mesial side.

Row *BB* includes three cusps in MNHN.F.SNP 2 Ma. In MNHN.F.SNP 335 W, the cusps cannot be described because of the wear. Cusp *BB1* is sub-equal in size to cusp *B4* and takes place at the same level. Cusp *BB1* is placed right next to cusps *B2* and *B3*, slightly more mesial than the *B2–B3* notch. Cusp *BB2* is mesial to cusp *BB1* but slightly more lingual. This cusp is smaller and takes place lower than cusp *BB1*. Cusp *BB3* is the mesialmost of the tooth. This cusp is mesiodistally aligned with cusp *BB1*. A crest extends row *BB* and runs labially to mesially close the lingual basin.

*Comments on MNHN.F.SNP 14 FW*
Only the distal part of MNHN.F.SNP 14 FW is preserved, with cusps *A1*, *B1*, *B2*, *BB1*, and a part of cusps *A2* and *B3*. Since morphometry is not applicable, this specimen is referred to *Theroteinus rosieriensis* following the position of cusp *B2* in relation to cusps *B3* and *BB1*. MNHN.F.SNP 14 FW differs from other teeth described above by a less developed cusp *A1* and more developed cusp *B3*.

**Wear**
***Lower molariforms***
In MNHN.F.SNP 309 W, RBINS.RAS 62 FW, RBINS.RAS 74 FW, and RBINS.RAS 77 FW, all cusps are abraded by wear. The labial side of row *b* shows a large, concave surface of wear which extends from the distal extremity of cusp *b2* to cusp *b4*. It is difficult to say if this concavity was present before the wear occurred or not, but it shows traces of wear, like the sides of the basin. MNHN.F.SNP 487 W also shows wear on the entire surface of the tooth but several facets are present. Cusp *a1* shows a steep, distolabial facet. Cusp *a2* shows a steep, distal facet on its apex connected with a steep, distolingual facet on its lingual side. Cusp *b1* shows a horizontal facet. Cusp *b2* is partially truncated by a concave, shallow, labio-distolabial facet, which extends on cusp *b4*. The apex of cusp *b4* shows a horizontal, distal facet.

***Upper molariforms***
In MNHN.F.SNP 2 Ma, only the apices of the cusps are abraded by wear. In MNHN.F.SNP 335 W, the cusps are more strongly abraded and show several facets. Cusp *A1* shows a steep, distal facet. Cusp *A3* shows possibly a steep, mesial facet. Cusp *B2* shows a shallow, mesiolabial facet. Other cusps of row *B* show one steep, mesial facet. Row *BB* shows one steep, mesial facet. In MNHN.F.SNP 14 FW, the wear seems more considerable. Cusp *A1* shows a shallow, distal facet. Cusp *A2* shows a large, horizontal, labial facet. Cusp *B1* shows a horizontal distal facet. Cusp *B2* shows a horizontal facet. Cusp *B3* possibly shows a shallow, mesial facet but is partially broken. Cusp BB1 shows a shallow, mesiolabial facet.

**Reconstruction of the dental row of *Theroteinus***
In such a poorly known group such as Haramiyida, the reconstruction of the dental rows from isolated teeth is notoriously difficult. Although five genera with complete or partial dentitions have been discovered in the last twenty years (*Jenkins et al., 1997*; *Zheng et al., 2013*; *Zhou et al., 2013*; *Bi et al., 2014*), there is no comparative study to

provide elements on interspecific and ontogenetic variations. The reconstruction of the dental row of *Theroteinus* is complicated by two additional problems: (i) the small number of specimens (n = 20) which prevents to evaluate the intraspecific variations, and (ii) the absence of premolariform specimens.

In upper molariforms, the variations of the development of cusps *A1* and *B1* and of the number of elements on the distolingual side of cusp *B2* can be related to the tooth position but also to individual or ontogenetic variations.

In lower molariforms, three specimens show characters possibly related to tooth position. MNHN.F.SNP 61 W has cusp *b2* more distal in comparison with cusp *a1* than other specimens. The first molar of *Haramiyavia* shows a similar character which may be a clue for a more mesial position in the dental row (*Jenkins et al., 1997*; *Luo et al., 2015*). MNHN.F.SNP 487 W has row *b* less high than in other specimens, cusp *b2* is especially much smaller in comparison with cusp *a1*. This difference of height is present in premolariforms of some haramiyids such as *Thomasia* as well, and it may consequently be a clue for a more mesial position in the dental row. However, MNHN.F.SNP 487 W does not show the distal shift of cusp *b2* seen in MNHN.F.SNP 61 W and the difference of height may also be related to ontogenetic variations. MNHN.F.SNP 497 W shows a distally reduced row *b*, especially cusp *b4*. Since this specimen does not show characters of the two other specimens, this reduction of row *b* may be a clue for the last locus in the dental row. Indeed, this locus displays often a partial reduction of the crown in other groups of mammaliaforms (see *e.g.*, *Debuysschere, 2016*). Since the reduction of row *b* could modify the occlusal function of the tooth, this interpretation may imply either that the last upper locus displays an equivalent reduction, or that this part of the tooth does not occlude with opposite teeth (*i.e.*, a more mesial position of the last upper locus).

## COMPARISONS

### Identification of *Theroteinus* species

The reappraisal of *Theroteinus nikolai* and the erection of *Theroteinus rosieriensis* sp. nov. are based on morphometric and morphologic characters.

### *Morphometry*

Measurements of the *Theroteinus* material are presented in Table 1 and descriptive statistics in Table 2. Because of the small number of upper molariforms (n = 4), no statistical test can be made to support this discussion. Statistical tests are possible on lower molariforms, but they need to be interpreted with caution because of the limited number of specimens available (n = 12). Means have been compared by the Welch's *t*-test, which is a variant of the Student's *t*-test (command 't.test()' in R software). This test assumes that data are normally distributed. This hypothesis has been tested by the Shapiro-Wilk test (command 'shapiro.test()' in R software), without rejection of the null hypothesis (Table S2). Since there are multiple comparisons, p-values have been corrected by the Holm-Bonferroni method (command 'p.adjust(method = "holm")' in R software) in order to control the family-wise error rate. The results of the *t*-test are

presented in Table 3. Graphically, the Fig. 7A shows two sets of upper teeth which do not overlap either by length or by width, and two sets of lower teeth which slightly overlap. Figure 7B shows that the same sets are present in length/width ratio, but with a more considerable overlapping between sets of lower teeth. Since differences of means in length, width, and length/width ratio are statistically significant (Table 3), specimens are divided between elongated small teeth and stocky large teeth.

### Morphology

In upper molariforms, two sets can be defined by the position of cusp *B2* which is either mesiodistally aligned with cusps *B3* and *B4*, or lingually shifted to face the lingual basin. In lower molariforms, two sets can be defined by the position of cusp *b4* which is either aligned with row *b*, or lingually shifted to face the saddle. Both of these variations are related together, because of the occlusal pattern. Indeed, cusp *B2* occludes lingually to cusp b4, consequently the latter cannot be shifted lingually if the former is not shifted either. Table 4 presents other morphological differences between sets defined above. However, in the current stage of knowledge, it is difficult to say if these differences are related to taxonomic, ontogenetic or individual variations.

Evaluating if observed differences are individual, populational, ontogenetic, sexual, or systematic variation is a hard question; especially with haramiyids for which these different variations, except systematic, are *terra incognita* for now. Such a question will be addressed when an adequate material is discovered. In this framework, it is more careful and parsimonious for the taxonomy to consider the most important observed differences as systematic variations, as do other studies (*e.g.*, *Bi et al., 2014*).

The sets defined by morphological and morphometric characters perfectly match. The lower and upper molariforms are associated following characters presented above and the two sets are considered as two species of the genus *Theroteinus*. The set including MNHN. F.SNP 78 W (defined as holotype by *Sigogneau-Russell, Frank & Hemmerlé, 1986*) is identified as *Theroteinus nikolai* and the second set is identified as *Theroteinus rosieriensis* sp. nov. Since the new hypodigm of *T. nikolai* includes all specimens referred to *Theroteinus* sp. described by *Hahn, Sigogneau-Russell & Wouters (1989)*, they are attributed to *Theroteinus nikolai*.

## Comparisons with other haramiyids

*Theroteinus* differs from all other known haramiyids by low and massive cusps, separated by very shallow notches and by short and narrow basins in comparison with the size of the tooth. This genus differs also by a small number of cusps in each row, especially only two cusps in row *a* (character seen only in some specimens of *Thomasia*).

*Theroteinus* possibly shares the presence of a supplementary upper lingual row *BB* with *Eleutherodon Kermack et al., 1998* (Middle Jurassic, England), *Megaconus* (Middle Jurassic, China), and *Millsodon Butler & Hooker, 2005* (Middle Jurassic, England). However, recognizing this similarity depends on the different interpretations of the specimens concerned, especially on the orientation of the teeth.
**Table 4 Summary of differences between lower and upper molariforms of *Theroteinus nikolai* and *Theroteinus rosieriensis* which are not included in diagnoses.**

|  | *T. nikolai* | *T. rosieriensis* |
|---|---|---|
| Lower teeth | - a vertical, weak medial ridge in the middle of the labial side of cusp *a2*<br>- alignment of the long axes of cusps *b3* and *b4*<br>- a lingual carina on cusp *b4* | - a very high saddle<br>- cusp *a2* twice times smaller than cusp *a1*<br>- sub-equality of the slopes of lingual and labial sides of cusp *a2*<br>- a mesiodistally less extended base of cusp *b2*<br>- cusp *b3* much smaller than cusp *b1*<br>- cusp *b3* slightly lingual to cusp *b2* |
| Upper teeth | - mesiodistal alignement of three cusps *A*<br>- a small cusp under the labial side of cusp *A3*<br>- a distally curved semi-circular cusp *B1* | - cusp *A1* slightly longer than cusp *A3*<br>- a concave surface on the mesiolingual side of cusp *A2*<br>- four cusps in row *B*<br>- cusp *B1* mesiodisally aligned with the saddle<br>- cusp *B3* more labial than cusp *B2*<br>- three cusp in row *BB*<br>- cusp *BB1* slightly more mesial than *B2–B3* notch |

Following the orientation of upper molariforms of *Eleutherodon* proposed by *Kermack et al. (1998)*; *Butler (2000)* named the labial row *A*, the middle row *B*, and the lingual row *BB*, which corresponds to the pattern of *Theroteinus*. However, *Meng et al. (2014*: p. 29)* proposed a second interpretation based on the comparison of the wear pattern of *Eleutherodon* with the wear pattern of *Arboroharamiya*. In this second interpretation, the labiolingual axis is inverted (*Meng et al., 2014*: Fig. 13). Although *Meng et al. (2014)* did not explicitly explain how they named the rows, it seems that they considered row *A* of *Butler (2000)* as row *B*, row *B* as row *A* and row *BB* as supplementary elements on the labial side of the tooth. According to the interpretation of *Meng et al. (2014)*, *Eleutherodon* does not share the presence of row *BB* with *Theroteinus*.

*Zhou et al. (2013)* did not name the rows of cusps of upper teeth of *Megaconus*. However, since the ultimate tooth shows only two rows, it is more parsimonious to consider these rows as rows *A* and *B*, which implies that the third lingual row present in the two previous teeth would be a row *BB*. This interpretation is consistent with the few published comments on the occlusion of *Megaconus* such as that of *Zhou et al. (2013*: Supplementary Information: p. 6): '[l]ower molars have two multicusp rows that alternately occlude in the two valleys between the three rows of cusps of M1 and M2.' However, *Meng et al. (2014)* questioned the orientation of the upper dentition of *Megaconus*. They proposed a reversal of the labiolingual axis and seemed to consider the labial row as a row *AA* (*Meng et al., 2014*: Fig. 14). It is difficult to decide between both interpretations upon the available data. It must be emphasized that both orientations are given with few details on the definition of rows and on the relationships between them. In this framework, several interpretations are possible, which prevents to conclude on the presence of row *BB* in *Megaconus* molariforms.

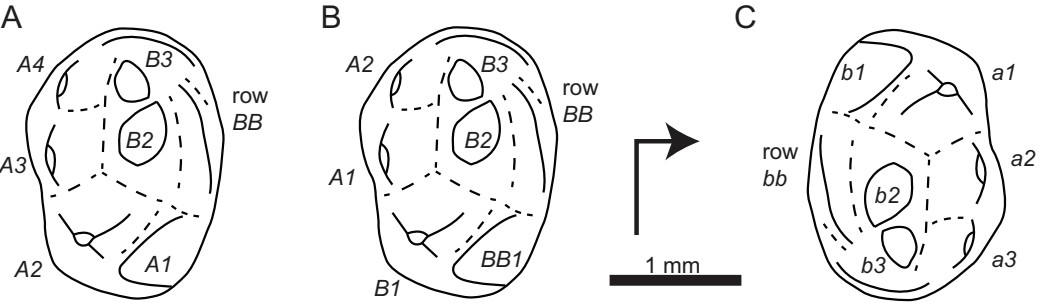

**Figure 8** Sketch drawings of specimen BDUC J 3 referred to *Millsodon* (Middle Jurassic, England), after *Butler & Hooker (2005*: Fig. 3C*)*. (A) Interpretation of *Butler & Hooker (2005)*; (B) Interpretation of *Hahn & Hahn (2006)*; (C) Interpretation proposed here. Right-angled arrow indicates mesial extremity and lingual side. Letters in italics correspond to cusp nomenclature.

The comparison with *Millsodon* is based on a specimen BDUC J 3, which is considered as a probable upper molar by *Butler & Hooker (2005*: p. 192*)*. If this interpretation is accepted, this specimen shows a row *BB* as *Theroteinus*, but it differs strongly from the latter by relationships of size and position between other cusps. Indeed, the pattern of cusps is very peculiar to an upper molariform and the two published interpretations of the specimen are very different from each other (*Butler & Hooker, 2005 contra Hahn & Hahn, 2006*) (Figs. 8A and 8B). The sole argument supporting the interpretation of BDUC J 3 as an upper molariform is the presence of a third cusps row. The rest of the crown looks more like a lower molariform, and can be described as follows: (i) a first row of cusps including a cusp much larger than others, (ii) a second row of cusps, which are similar in size with small cusps of the first row, (iii) in the second row the largest cusp is close to the large cusp of the first row but not labiolingually aligned with it, and (iv) a cusp located at one extremity of the tooth, aligned with the second row, but separated from it by the large cusp. No one other haramiyid upper tooth matches this pattern, unlike lower molariforms of *Thomasia* and *Haramiyavia*. This new interpretation of specimen BDUC J 3 as a lower molariform (Fig. 8C) is favoured here and implies two consequences. First, the referral of specimen BDUC J 3 to the genus *Millsodon* needs a reassessment. Second, the specimen can be compared with the lower molariforms of *Theroteinus*. Several characters are shared by these teeth: (i) the presence of few cusps by row, (ii) low and obtuse cusps, (iii) and a short and narrow basin. Moreover, the third row of BDUC J 3, which is labial in this interpretation, can be considered as development of the labial bulge present in some specimens of *Theroteinus* (Figs. 4B, 4C and 5). However, new examinations of BDUC J 3 would be necessary to discuss further these points, which is impossible as this specimen is lost according to *Butler & Hooker (2005*: p. 191*)*.

## DISCUSSION

*Theroteinus* is referred to Haramiyida because of the presence of parallel rows of cusps. Moreover, its molariforms show a pattern of cusp rows *a/A* and *b/B,* in size and relative position of cusps, which is strongly similar to patterns seen in *Thomasia* and *Haramiyavia*.

In addition, the occlusal pattern of *Theroteinus* is similar to the pattern of *Thomasia* with row *B* occluding into the lower basin. However, *Theroteinus* is very peculiar among haramiyids. The genus is defined by characteristic morphological characters (see above) and by a different masticatory movement. Indeed, *Theroteinus* is the only haramiyid for which the wear pattern does not highlight a horizontal movement of the jaw during mastication (*Sigogneau-Russell, Frank & Hemmerlé, 1986* and see above). Such a wear pattern and the small size of the basins support an essentially vertical masticatory movement. Because of these differences, *Theroteinus* has occupied since a long time a special place in the systematics of haramiyids, either as sister-group of the whole order Haramiyida (*Hahn, Sigogneau-Russell & Wouters, 1989*) or isolated in a sub-order (*Butler, 2000*; *Hahn & Hahn, 2006*). In the absence of a relevant cladistics analysis including *Theroteinus*, the suborder Theroteinida is conservatively used in order not to complicate the taxonomy of haramiyids, which has already seen many changes. In the same purpose, the name 'Theroteinida' is used unchanged although it would be best to modify it. As underlined by *Hahn & Hahn (2006*: p. 189*)*, the suffix of the name of a sub-order should be different from the suffix of the name of the including order. However, the suffix '-ina' suggested by *Hahn & Hahn (2006*: p. 189*)* cannot be used since it is reserved for the name of a subtribe by article 29.2 of the ICZN (*International Commission on Zoological Nomenclature, 2000*).

The only taxon closely related to *Theroteinus* is *Millsodon*, which is considered as a member of Theroteinidae by *Hahn & Hahn (2006)*. *Butler & Hooker (2005*: p. 192*)* compared the upper tooth of *Millsodon* with the upper molariforms of *Theroteinus* and suggested that *Millsodon* could be 'a derivative of the Theroteinidae or a specialized relative of the Haramiyidae.' However, *Butler & Hooker (2005)* considered *Millsodon* as indeterminate at familial rank and did not compare its lower molariforms with lower molariforms of *Theroteinus*. *Hahn & Hahn (2006)* considered that lower molariforms of *Millsodon* can be derived from lower molariforms of *Theroteinus*. This interpretation is based on specimen MNHN.F.SNP 226 W. *Hahn & Hahn (2006*: p. 184*)* considered that differences between this specimen and other lower molariforms of *Theroteinus* cannot be explained only by wear and that this specimen represents 'a new taxonomical unit (perhaps a genus and a species)' and is intermediate between *Theroteinus* and *Millsodon*. This interpretation is questionable. First, the specimen MNHN.F.SNP 226 W is very poorly preserved, not only because of wear during life but also probably because of taphonomic processes. Creating a new genus and outlining an evolutionary scenario only on the base of such a poorly preserved specimen is problematic. Second, comparisons between *Theroteinus* and *Millsodon* are difficult. On the one hand, the description of the upper tooth of *Millsodon* is questionable (see above). On the other hand, all lower teeth of *Millsodon* are heavily worn (*e.g.*, *Butler & Hooker, 2005*: Figs. 1D and 1E), and the cusps are difficult to characterize. However, all specimens of *Millsodon* clearly show a well-developed basin, which is distinct from *Theroteinus*. Consequently, the family Theroteinidae is considered here as monogeneric.

## INSTITUTIONAL AND OTHER ABBREVIATIONS

**BDUC**   Biology Department, University College, London, United Kingdom
**MNHN**   Muséum National d'Histoire Naturelle, Paris, France
**RAS**    Rosières-aux-Salines, another name for the study site
**RBINS**  Royal Belgian Institute of Natural Sciences, Brussels, Belgium
**SNP**    Saint-Nicolas-de-Port.

## ACKNOWLEDGEMENTS

This study is based on the author's Ph.D. thesis work at the MNHN (doctoral school 'ED 227, Sciences de la Nature et de l'Homme'), supervised by Emmanuel Gheerbrant and Ronan Allain. The author thanks the following people: Pascal Godefroit for his help during several visits to the RBINS; Alexandre Lethiers for helping with the preparation of the drawings; Julien Cillis for taking the SEM photographs in the RBINS; Miguel Garcia Sanz for his work on the AST-RX platform 'Plateforme d'accès scientifique à la tomographie à rayons X' supervised by the UMS 2700 'outils et méthodes de la systématique intégrative CNRS-MNHN' as well as Florent Goussard and Damien Germain for their help in the processing of 3D images; and Ronan Allain, Emmanuel Gheerbrant, Thomas Martin, Mark Young, and an anonymous reviewer for their critical reading of the manuscript.

### Funding

This study has been supported by the ATM 'Biodiversité actuelle et fossile. Crises, stress, restaurations et panchronisme: le message systématique', the ATM 'Emergences', and by the UMR 7207 'Centre de Recherche sur la Paléobiodiversité et les Paléoenvironnements'. The funders had no role in study design, data collection and analysis, decision to publish, or preparation of the manuscript.

### Grant Disclosures

The following grant information was disclosed by the authors:
ATM 'Biodiversité actuelle et fossile Crises, stress, restaurations et panchronisme: le message systématique', the ATM 'Emergences', and by the UMR 7207 'Centre de Recherche sur la Paléobiodiversité et les Paléoenvironnements'.

### Competing Interests

The author declares that they have no competing interests.

### Author Contributions

- Maxime Debuysschere conceived and designed the experiments, performed the experiments, analyzed the data, contributed reagents/materials/analysis tools, wrote the paper, prepared figures and/or tables, reviewed drafts of the paper.

## Data Deposition

The raw data is included in the manuscript.

## New Species Registration

The following information was supplied regarding the registration of a newly described species:

Theroteinus rosieriensis sp. nov. LSID: urn:lsid:zoobank.org:act:F3C6B3B3-1733-4625-942F-9C085A51116A.

Publication LSID: urn:lsid:zoobank.org:pub:57401966-D5B5-468C-94FD-115C0C32FE00.

## Supplemental Information

Supplemental information for this article can be found online at http://dx.doi.org/10.7717/peerj.2592#supplemental-information.

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
