# Peer review of "A reappraisal of Theroteinus (Haramiyida, Mammaliaformes) from the Upper Triassic of Saint-Nicolas-de-Port (France)"

_PeerJ, doi:10.7717/peerj.2592_

## Round 0.1 · original submission · Minor Revisions

Dear author,

I have accepted the decision of ‘minor revisions’ from the two reviewers.

I have some additional comments that the authors should also address prior to resubmission:

1. In the title, please ensure Upper and Triassic are capitalised (they aren’t on the system, but are in the MS itself).
2. ‘Primitive’, ‘basal’ would be more appropriate.
3. In your diagnoses, can you indicate which characteristics are autapomorphic.
4. Can you add in a section explaining why the new species is not an ontogenetic morph, or exemplar of individual/population/sexual variation, of the type species?
5. There are some typos in the text and tables that will require correcting prior to re-submission. Please note PeerJ does not provide a full language proofing system, thus it is up to the author to ensure spelling and grammar are free of errors.
6. As you are making multiple comparisons within the same dataset (n=3), you will need to correct your alpha value (i.e. 0.05 is too low). A Bonferroni correction would be suitable.
7. In table 2 you give three descriptive statistics (mean, standard deviation, and median). Can you state what distribution the data have?

·

Basic reporting

The Article is well written but the English needs some polishing. I have made some suggestions in the attached annotated manuscript, but not being a native Speaker myself, I suggest the English of the Article to be checked by a native Speaker.

Experimental design

No comments

Validity of the findings

No Comments

Additional comments

This is a well-written manuscript that deals with a poorly known Group of basal mammaliaforms. The descriptions are detailed and clear, and the conclusions and interpretations are Sound. The description and Interpretation of this material is an important contribution to early mammal Research.
I have attached an annotated manuscript file with some linguistic/orthographic corrections and few comments on clarity.

Reviewer 2 ·

Basic reporting

This study revisits the material of Theroteinus nikolai and named a new species of the genus. It provides additional information to the otherwise poorly known genus of the phylogenetically controversial Haramiyida. The new data are certainly welcome and publishable, but I have some concerns about the presentation and interpretation of the manuscript, as I listed below. In addition, I have made many minor comments in the pdf file and the marked ms is attached to the author for consideration. Also, I think the English can be further improved.

Experimental design

No comments.

Validity of the findings

No comments.

Additional comments

In addition to those marked on the pdf text, I have the following comments for the author:

Line 136: In the Dental nomenclature, it’s better the author specifies the lingual and labial (buccal) side of the tooth when a/A, b/B cusp rows are referred.

It will be clearer if the author explain how the teeth are identified as molariform, not premolariform; how the left and right teeth are identified; and how the lingual and labial cusp rows are identified for any upper and lower molariform. The criteria for identification of these isolated teeth should be addressed.

Line 166 – The author said: “In lower molariforms, the lingual row a includes the largest cusps. The cusp a1 is especially much larger than the others cusps and it is located on the mesiolingual side of the crown.” Why do the author think this is the case? In several other studies, particularly those on eleutherodontids, the largest cusp on the lower molariform is b2. The author should provide an argument to explain why does he/she think the largest cusp of the lower molariform is the mesiolingual (a1) instead of mesiolabial (b2).

Line 391 - The type locality must be specified when you name a new species. You need to have “Type locality and horizon: Saint-Nicolas-de-392 Port (Upper Triassic, France)” as part of the formal proposal for a new species.

Lines 672-673: The author stated: “Moreover, its (Theroteinus) molariforms show a pattern of cusps in size and relative position which is strongly similar to patterns seen in Thomasia and Haramiyavia.” This statement cannot convince me if the interpretation of the teeth (mainly the upper molariforms) of Theroteinus is correct. As described, the molariform of Theroteinus has “supplementary cusps” on the lingual side of the upper molariform, but in Haramiyavia, the “supplementary cusps” are C-cusps, which are on the labial side of the upper molariform (labial to A cusps). The molariforms of Haramiyavia are in situ so that their orientation should have no problem. If the tooth orientation for Theroteinus is correct, as the author interpreted, then the upper molariforms of Theroteinus and those of Haramiyavia are extremely different – the former has extra cusps added on the lingual side of the upper molariform, whereas the latter has the extra cusps added on the labial side of the molariform. However, if the orientation of the tooth is reversed, such as the holotype MNHN.F.SNP 78 W being identified as a left, NOT a right, upper molarifrom, I’d agree with the author’s statement listed above.

Annotated reviews are not available for download in order to protect the identity of reviewers who chose to remain anonymous.

---

## Round 0.2 · accepted · Accept

Dear author,

Many thanks for your revised manuscript. After reading it, and the decision from one of the reviewers, I have accepted it for publication in PeerJ.

Once again, thank you for submitting your manuscript to PeerJ and I hope you will use us again as your publication venue.

If we need to clarify any details required to move the manuscript forward, then our production staff will get in touch with you. Otherwise, a proof will be forthcoming shortly for your review.

Congratulations and thank you for your submission.

Reviewer 2 ·

Basic reporting

I have read the revised version of the manuscript: "A reappraisal of Theroteinus (Haramiyida, Mammaliaformes) from the Upper Triassic of Saint-Nicolas-de-Port (France)"by Maxime Debuysschere, and think the author has addressed some of the minor issues I raised in the earlier version of the ms. I think the work is ready to go for publication.

Experimental design

No comments.

Validity of the findings

No comments

Additional comments

No more comments.